# Tuberculosis in Adolescents and Young Adults: Emerging Data on TB Transmission and Prevention among Vulnerable Young People

**DOI:** 10.3390/tropicalmed6030148

**Published:** 2021-08-05

**Authors:** Katherine M. Laycock, Leslie A. Enane, Andrew P. Steenhoff

**Affiliations:** 1Division of Infectious Diseases, Department of Pediatrics, Children’s Hospital of Philadelphia, Philadelphia, PA 19104, USA; steenhoff@chop.edu; 2The Ryan White Center for Pediatric Infectious Disease and Global Health, Department of Pediatrics, Indiana University School of Medicine, Indianapolis, IN 46202, USA; lenane@iu.edu; 3Department of Pediatrics, Perelman School of Medicine, University of Pennsylvania, Philadelphia, PA 19104, USA; 4Global Health Center, Children’s Hospital of Philadelphia, Philadelphia, PA 19146, USA

**Keywords:** screening, transmission, schools, adherence, retention in care, youth-friendly services

## Abstract

Adolescents and young adults (AYA, ages 10–24 years) comprise a uniquely important but understudied population in global efforts to end tuberculosis (TB), the leading infectious cause of death by a single agent worldwide prior to the COVID-19 pandemic. While TB prevention and care strategies often overlook AYA by grouping them with either children or adults, AYA have particular physiologic, developmental, and social characteristics that require dedicated approaches. This review describes current evidence on the prevention and control of TB among AYA, including approaches to TB screening, dynamics of TB transmission among AYA, and management challenges within the context of unique developmental needs. Challenges are considered for vulnerable groups of AYA such as migrants and refugees; AYA experiencing homelessness, incarceration, or substance use; and AYA living with HIV. We outline areas for needed research and implementation strategies to address TB among AYA globally.

## 1. Introduction

Tuberculosis (TB), the leading infectious cause of death globally by a single agent before the emergence of SARS-CoV-2, continues to challenge prevention and care efforts worldwide. Attention has long focused on TB cases among adults, and surveillance data have not been disaggregated to allow for estimates of TB among adolescents. Consequently, prevention and care policies typically overlook adolescents. A recent estimate calculated that 1.8 million adolescents and young adults (AYA, ages 10–24 years) developed TB disease in 2012, representing 17% of all new TB disease diagnoses globally [1]. Moreover, adolescence encompasses a period of critical change that combines increasing susceptibility to TB with unique developmental and social characteristics to create particular risks for TB transmission, infection, and disease [2,3,4]. Successful strategies to end TB for all thus require TB prevention and care approaches that meet adolescents’ needs.

This narrative review aims to summarize the current knowledge on TB transmission dynamics among AYA, with a focus on needed strategies to address the TB epidemic among AYA. To contextualize TB transmission, we distinguish between “TB infection” and “TB disease”. TB infection (often called “latent TB infection”) indicates an asymptomatic or transiently mildly symptomatic stage of immune sensitization to *Mycobacterium tuberculosis*. Individuals with TB infection (without progression to active disease) are unlikely to transmit TB to others. In adults with TB infection, 10% will progress from TB infection to develop TB disease during their lifetime. The risk of developing TB disease is increased in adults with immune suppression and is also higher in children due to their developing and not yet fully functional immune systems [2]. TB disease (often called “active TB”) includes a range of clinical manifestations—which may be pulmonary, extrapulmonary, or both—that develop once infection with *M. tuberculosis* progresses to invasive disease. Individuals with pulmonary TB disease can potentially transmit *M. tuberculosis* to others via exhaled airborne particles that are inhaled by others.

To consider the important components of physiologic, developmental, and social maturation that extend into young adulthood, in this review, we include AYA aged 10–24 years. We describe vulnerable groups of AYA who face increased risk for poor outcomes. We examine major settings for TB transmission to and among AYA and discuss best screening practices for identifying AYA with TB. Finally, we highlight priority areas to strengthen TB prevention, case-finding, and care among AYA.

## 2. Methods

For this narrative review, we searched PubMed using the MeSH term “tuberculosis” and key word “adolescent” to identify original research titles published in English between 2016 and April 2021. We examined abstracts and full text articles to identify relevant studies that provided disaggregated data regarding individuals between 10 and 24 years of age (inclusive of narrower age groups). Additional queries combining the MeSH term “tuberculosis” with other key words such as “school”, “migrant”, or “military” were performed to maximize yield of relevant literature on settings of TB transmission among AYA. For key populations with limited published TB data specific to AYA (e.g., migrant AYA), we also included studies encompassing broader age groups to attempt to provide more insight into specific vulnerabilities of AYA from these populations. We hand-searched the references of all selected titles to identify additional relevant titles. Included studies were reviewed in detail, with findings summarized and synthesized in this narrative review.

## 3. Epidemiology of Adolescent TB

The true burden of TB among AYA is unclear. Reporting classifications used by the World Health Organization (WHO) have historically omitted the adolescent age group by categorizing TB cases as “children” (0–14 years) or “adults” (15 years and older) [5]. Data on AYA with TB thus come largely from single-center or national studies, with cases often disaggregated into varying age groups. A recent study estimated 1.8 million (uncertainty interval 1.23–3.00 million) incident cases of TB disease among AYA globally in 2012, representing approximately 17% of all new TB cases [1]. Despite challenges quantifying the burden of TB among young people, adolescence has emerged as a distinct period of risk.

### 3.1. Prevalence, Gender, and Risk Factors for AYA TB

In settings with high TB incidence, the prevalence of TB infection increases throughout adolescence as individuals accumulate exposures to *M. tuberculosis* [6]. The risk of progression to TB disease, which is lowest among children aged 5–9 years, also increases throughout adolescence [2]. Before the availability of anti-TB chemotherapy and the emergence of HIV, the likelihood of progression from TB infection to TB disease was estimated to increase from 3.8% in children aged 5–9 years to 6.4% at ages 10–14 years and 10–13% at ages 15–24 years, with most of these children and AYA developing TB disease within one year of infection [7]. In the modern era, variable reporting methodologies have complicated attempts to quantify the global incidence of TB infection and the risk of progression to TB disease among AYA [1,2]. One model estimated that in 2014, at least 20% of young adults in some regions developed TB infection [8]. Another recent global estimate calculated that in 2012, TB disease developed in 192,000 young adolescents aged 10–14 years, 535,000 older adolescents aged 15–19 years, and 1,049,000 young adults aged 20–24 years worldwide [1].

Age-related trends in immunologic susceptibility persist among AYA across regions [2,9,10,11]. Puberty is associated with changes in immunity that may contribute to an increased risk of progression to TB disease among AYA [2,9,10,11]. Disease type shifts during adolescence from the less transmissible paucibacillary disease characteristic of younger children to highly transmissible forms of TB, including cavitary lung disease and laryngeal TB, typically seen in adults [2,9,10,11].

Nuanced sex differences in susceptibility to TB appear during adolescence [2,10]. Though TB affects younger girls and boys equally, the TB risk for females increases around the time of menarche, resulting in a higher TB incidence and a higher risk of disease progression when compared to age-matched male adolescents [2,10]. Increased TB risk for females relative to males appears to peak at mid-adolescence [2,10]. Sex differences are also seen in TB-related mortality, which may intersect with higher vulnerability to HIV among female AYA [2,10,12]. A study in Cape Town, South Africa, a region with high TB and HIV prevalence, found that of adolescents aged 15–19 years with drug-susceptible TB, female adolescents had more than twice the risk of mortality than male adolescents [12]. Among young adults and older individuals, the higher TB risk shifts to males [2,10]. Reasons for these shifting sex differences remain unclear but may reflect a combination of factors [2,10,12]. Proposed influences include factors increasing susceptibility to TB among female adolescents (due to physiologic changes at menarche, a higher incidence of HIV infection among female AYA, TB risks associated with pregnancy, and/or other drivers yet to be identified) or factors resulting in undiagnosed TB among male adolescents, who might be less likely to access care to receive a TB diagnosis and/or may face higher mortality from interpersonal violence or other noninfectious causes [2,10,12].

AYA share many of the same risk factors for TB infection and progression to disease as older adults (Table 1) [2,3,5]. Some predisposing comorbidities may occur less frequently among AYA than among adults. For example, the global incidence of diabetes mellitus in AYA is estimated to be less than half that seen in many older age groups [13]. However, rates of obesity and youth-onset type 2 diabetes mellitus have surged among AYA in some regions, and future increases are predicted among AYA worldwide [14,15]. Providers may address many individual risk factors as part of routine TB care, though some providers may hesitate to broach subjects such as drug and alcohol use with their AYA patients [16].

### 3.2. TB Preventive Treatment Regimens for AYA

For individuals with TB infection and those at high risk for TB disease, TB preventive treatment (TPT) can stop progression to TB disease, thus reducing TB transmission [17]. Monotherapy with isoniazid (often called “isoniazid preventive therapy” or IPT) for at least six months formed the basis of TPT for many years [18]. New, shorter rifamycin-based medication regimens, which do not appear to increase adverse effects or decrease efficacy, now offer alternatives for many individuals (Table 2) [18].

These shorter regimens may be more acceptable to AYA than isoniazid monotherapy, thus supporting treatment adherence and completion [19,20,21]. Providers selecting a TPT regimen should consider local TB resistance patterns and the potential for drug-drug interactions between rifamycins and other medications, such as antiretroviral therapy (ART) and hormonal contraceptives [18]. Anticipatory guidance for AYA, as for individuals of all ages, should include counseling on alcohol avoidance while taking any TPT regimen [18].

### 3.3. TB Disease Treatment for AYA

For AYA with TB disease, treatment requires prolonged adherence to multiple medications. These treatment regimens, while identical to those used for adults with TB disease, can create unique adherence challenges for AYA and should be provided as part of comprehensive services tailored to AYA needs [3,22,23,24]. Approaches to treatment of TB disease for AYA are beginning to be defined and are considered in detail elsewhere [3]. To date, global TB programs have not implemented adolescent-friendly TB services or adolescent-specific strategies to support adolescents in the completion of TPT or treatment of TB disease [22,23]. Research and dedicated implementation strategies for adolescent TB treatment are urgently needed.

### 3.4. Outcomes for AYA

The complex treatment regimens necessary to treat TB infection and disease present particular challenges for AYA. Adolescent development is characterized by drivers such as identity formation, building peer relationships, and increasing independence from caregivers, and includes important tasks for educational attainment and entering work life [3,24]. Perceived TB stigma and social isolation—which disrupt AYA needs for education, socialization, and occupations (at later ages)—add further challenges. Disruptions may be especially acute for AYA with drug-resistant TB, who may undergo prolonged enforced isolation or hospitalization [22,25,26]. As a result, AYA with TB infection and disease experience higher rates of loss to follow-up (LTFU) than either younger children or older adults [12,27,28,29,30]. A study in Botswana found that adolescents aged 10–19 years were lost from TB care more often than young adults aged 20–24 years and at twice the rate of older adults [27]. In two different communities in South Africa, LTFU occurred more frequently among older AYA, with rates of LTFU highest among young adults aged 20–24 years [28,30]. Individuals of any age with incomplete TB treatment can develop drug resistance and, if sputum smear-positive, may contribute to ongoing TB transmission [31].

### 3.5. Vulnerable AYA Populations

Certain groups of marginalized AYA may experience more pronounced TB risks, resulting from increased likelihood of acquiring TB, complex barriers to accessing and engaging in TB care, and/or immunologic susceptibility to TB. These vulnerable populations include migrant AYA; AYA living with HIV; and AYA experiencing homelessness, substance abuse, or incarceration (Figure 1).

#### 3.5.1. Migrant AYA

Migrant AYA, including refugees and other displaced AYA, have heterogeneous experiences that are poorly captured in surveillance data, complicating efforts to measure their TB risk and outcomes [32,33,34,35,36]. However, displaced people of all ages often face a disproportionately high risk for TB due to a combination of overcrowding, malnutrition, poverty, psychological stressors, and healthcare system disruptions that lead to delayed diagnosis and treatment [32,37,38]. Birth in a country with high TB prevalence consistently predicts an increased risk for TB infection and disease [32,37]. Risk also increases for migrant AYA who have longer transit times from their country of origin and who spend time in informal settlements (e.g., camps, detention centers), which may act as bottlenecks that support TB transmission [32,34,35,36,37,38,39]. AYA living in cross-border regions, such as Papua New Guinea and the United States-Mexico border, experience additional systems challenges while navigating semipermeable borders that can prevent equitable access to TB care and compound their risk [40,41,42]. With numerous barriers to TB care, migrant AYA also experience higher rates of LTFU and other unfavorable outcomes than resident AYA [34,37,40].

#### 3.5.2. AYA Living with HIV

Though the incidence of new HIV infections in AYA has declined overall, progress lags behind global targets and remains uneven, with AYA from key populations and in some regions experiencing more limited progress [43]. In 2019, an estimated 3.4 million people aged 15–24 years were living with HIV worldwide [43]. AYA living with HIV (ALHIV) experience more complex barriers and remain more underserved in many areas of HIV care compared to other age groups [44]. Data on TB in ALHIV remain limited, but growing evidence demonstrates that ALHIV also experience excess TB risks [27,29,43,44,45,46,47,48,49,50,51]. Early initiation of ART reduces the risk of TB disease in ALHIV [45,46]. However, despite improved access to ART, ALHIV continue to have higher rates of TB infection and disease than HIV-negative AYA in the same communities [47,48]. When compared to their HIV-negative peers with TB, AYA with TB-HIV coinfection consistently experience higher rates of LTFU and death [27,29,48,49,50,51].

#### 3.5.3. AYA Experiencing Homelessness, Substance Use, or Incarceration

Adults who experience homelessness, harmful alcohol or drug use, or incarceration have documented excess risks for acquiring TB and for unfavorable TB outcomes, but very little is known about AYA with the same vulnerabilities [52,53,54,55]. The same immunologic, socioeconomic, and health systems factors that impair the health of these adults likely also give vulnerable AYA excess TB risks.

Limited data show that these vulnerable AYA develop TB infection and disease at higher rates than other AYA. AYA who experience homelessness or incarceration likely have a higher risk for acquiring TB because they live in congregate settings that support TB transmission [55,56,57]. Data on TB incidence in AYA who use tobacco are minimal. Small studies conducted with school-aged children in Brazil and Mongolia suggest that AYA who smoke have a higher risk of developing TB infection and of progressing from TB infection to disease than their non-smoking peers [58,59]. AYA exposed to secondhand tobacco smoke also have an increased risk for TB infection and disease that, while smaller than the risk seen in younger children, likely exceeds that seen in older adults [60]. The incidence of TB among AYA who abuse alcohol or illicit drugs has not been defined but is likely high.

For these AYA, vulnerabilities likely also contribute an increased risk for unfavorable TB outcomes. One study of HIV-negative AYA with TB in Brazil found that AYA who had at least one vulnerability—including homelessness, incarceration, tobacco use, illicit drug use, or alcohol abuse—experienced unfavorable outcomes at a rate three times higher than their peers [51]. Excessive use of stimulants such as khat may also increase the risk for unfavorable outcomes [61].

## 4. Settings of Adolescent TB Transmission

AYA are highly mobile, with more social contacts each day than either young children or older adults [62,63]. While a new TB diagnosis in a child under age 5 years generally signals recent TB transmission within the household, the complex social networks of AYA, and the potentially longer interval between TB infection and onset of TB disease, complicate identification of the exact settings for TB transmission to AYA [2].

### 4.1. Household Exposures

For AYA, as for people of all ages, contact with a household member who has pulmonary TB disease increases the risk of TB infection and subsequent progression to disease. Household exposures contribute to ongoing community TB transmission to an unclear extent, particularly in regions with high TB prevalence [64,65]. However, the relative impact of household exposures appears smaller for AYA than for younger children, and this impact may decline throughout adolescence. An international meta-analysis of the risk of TB infection in children aged 0–14 years demonstrated that household contact conferred less additional risk of infection for children aged 10–14 years compared with children aged 0–4 years [66]. In Peru, a prospective study of household and community exposures found that the excess risk of exposure to smear-positive, culture-positive household members led to 58% of TB infections in children aged younger than 1 year, decreasing to 48% in children aged 10 years and 44% in children aged 15 years [67]. Studies in Uganda and South Africa found no significant association between household contact and risk of TB infection among older AYA [68,69]. In India, adolescents aged 15–18 years and people aged 19–45 years had a similar risk of TB infection after household exposure [70]. These findings highlight that for AYA, increasing mobility provides increasing opportunities for cumulative exposures outside the home.

### 4.2. School Exposures

Outside their homes, AYA tend to spend the majority of their time in school settings. Schools gather groups of AYA in repeat, prolonged close contact in buildings that may have poor ventilation, all conditions that favor TB transmission [63,71]. In communities with high TB prevalence, schools thus become significant sites of TB transmission. A modelling study in high-prevalence communities in Cape Town, South Africa, mapped adolescents’ exhaled CO_2_ levels during their typical daily activities at home and in the community [72]. Findings estimated that for adolescents aged 15–19 years, half of TB transmission occurred in schools [72]. With crowding, TB may become more prevalent in schools than in the larger community, as demonstrated by surveillance studies in Ethiopian universities [73,74]. Because AYA with pulmonary TB disease often have highly transmissible disease, schools also remain important sites of potential TB transmission in communities with low TB prevalence [75,76].

Outbreaks of TB occur in schools worldwide but, likely due to health resource constraints in high-prevalence settings, most reports of school-based contact investigations paradoxically come from low-prevalence communities [77,78]. Unsurprisingly, the risk of TB transmission among AYA in schools increases with increasing proximity and duration of exposure. Contact investigations that recorded classroom seating charts found that students who were routinely seated next to a student with TB had the highest incidence of TB infection [26,79,80]. School building designs that reduce shared air may conversely limit transmission, as with one outbreak that affected only a single classroom that opened directly to the outdoors and did not share a corridor with other classrooms [80]. Other investigations that noted the duration of time spent in contact with a student with TB have demonstrated that cumulative in-school contact for at least 8 h per week significantly increased other students’ risk of infection [81,82]. School-based transmission from an adult teacher to an adolescent student with fewer than 10 h of contact has also been reported [83]. Delayed discovery of active TB cases and delayed contact investigations feature in many school outbreaks, leading to extensive spread of infection [26,78,79,80,82,83,84,85].

Boarding school settings present an even greater risk for TB transmission than day school settings, likely due to increased opportunities for transmission in classrooms and dormitories. A case-control study of AYA with TB at over 200 boarding schools in China found that students who were both roommates and classmates of a student with pulmonary TB disease had a relative risk of TB infection more than three times higher than students who were only classmates [86]. Shared dormitory spaces have also been linked to a higher risk of developing of TB disease. In one genotypically linked outbreak at a boarding school in China, roommates of the index student had a significantly higher incidence of TB disease than the student’s other close contacts [85]. Similarly, active case finding at 11 boarding and day schools for Tibetan refugees in India identified higher incidence of both TB infection and disease in boarding schools and dormitories, compared with day schools and classrooms [87].

Schools that enroll students from broad catchment areas may face additional challenges in TB prevention. In one community in Uganda, TB infection was more prevalent among AYA who attended distant boarding schools than among AYA who attended community schools [69]. This additional risk may reflect exposures within the boarding schools as well as other exposures encountered during travel. Contact investigations in these schools may also necessitate close cross-jurisdictional communication between TB programs. One outbreak of extensively drug-resistant TB at a university in Romania affected students from five countries and required extensive international collaboration [88].

Attention to school-based transmission should not ignore factors that influence the risk of TB transmission in the surrounding community. Phylogenetic analysis of isolates from a school-based outbreak among AYA in Macau identified three separate clusters and several unrelated strains of TB, indicating a larger community source [89]. In South Africa, a prevalence survey at 8 high schools found that the risk of TB transmission at each school served as a marker of its students’ socioeconomic status [90]. Even in boarding school settings, TB enters from the community. A large case-control study of AYA at boarding schools in China found that index students had a history of significant household exposure but not prior classroom exposure [91]. Community surveillance studies in Uganda and Kenya also identified a higher prevalence of TB infection among AYA who had left school compared with school-going AYA, though only a small proportion of AYA in each study had left school [69,92,93,94].

### 4.3. Exposures on Public Transit

Public transit has emerged as an important contributor to TB transmission, particularly in urban high-prevalence communities. A case-cohort study conducted in Lima, Peru, found that commuting via minibus was an independent risk factor for TB disease [95]. The same modelling study that explored school-based transmission in Cape Town, South Africa, demonstrated that exposures on public transit held similar importance for TB transmission to AYA [72]. As with school-based exposures, the degree of exposure on transit appears to affect the risk of TB transmission, with prolonged close contact associated with higher risk of TB infection than more brief contact [96]. One CO_2_ monitoring study in Dar es Salaam, Tanzania, calculated that transit drivers had an annual TB infection risk 10 times that of transit riders, a result of longer time spent on poorly ventilated buses [97].

### 4.4. Other Community Exposures

Identifying other community settings of TB transmission has proved challenging for all age groups. Studies in high-prevalence areas have found varying degrees of TB transmission at indoor gathering sites such as markets and places of worship, highlighting the heterogeneity of these settings and the duration of time individuals spend in them [62,63,97]. However, reported outbreaks associated with social gathering sites illustrate the need to consider AYA social networks in a broad sense. One phylogenetically linked outbreak that affected people aged 19–28 years in Singapore spread at local area network gaming centers [98]. Another genotypically linked cluster that affected children, AYA, and older adults in the United Kingdom involved multiple venues including an internet café, barbershop, and football club [99].

### 4.5. Military Exposures

Given that most military recruits worldwide are AYA, TB transmission in military settings deserves specific mention. The prevalence of TB disease among military personnel is generally lower than among the corresponding general population, as demonstrated by surveillance of service members in India and the United States [100,101]. This is attributed to the “healthy warrior effect”, in which strict induction standards that exclude recruits with certain underlying immunocompromising conditions create a population with overall better health status [102]. However, outbreaks of TB continue to occur in military settings, with particular impact on AYA [100,103].

AYA in the military develop TB disease through either reactivation of preexisting TB infection or infection acquired after military enrollment. In countries with low TB prevalence, most military personnel with TB have TB infection at the time of their enrollment [100,104]. Circumstances of military service may place AYA at increased risk of acquiring TB, though this association has proved challenging to study [105]. Proposed reasons for an increased risk during training and deployment include residence in crowded communal settings (e.g., barracks, naval ships, submarines); exposures during military or humanitarian operations in high-prevalence communities, where conflict may have disrupted local TB programs; and stress-driven decreases in immune function that may increase both infectivity and susceptibility [102,106].

## 5. Identifying AYA with TB

Halting TB transmission among AYA hinges in large part on prompt diagnosis and treatment for individuals with TB infection and disease. Approaches vary widely around the world. However, regardless of local TB prevalence, most TB programs rely on passive case-finding—that is, identifying TB cases among AYA who either present to a healthcare facility with symptoms or are found in contact investigations—rather than active measures.

Broader use of active case-finding in high-prevalence settings can reduce TB transmission by facilitating prompt diagnosis and treatment for AYA. In the Russian Federation, for example, TB prevalence has fallen across all age groups with the implementation of numerous programmatic efforts including, since 2014, annual testing for TB infection for all children and adolescents under age 18 years [107]. More extensive use of TPT among AYA can also reduce TB transmission by preventing progression to TB disease. Ongoing efforts aim to identify the AYA at highest risk for TB through the development and validation of individualized risk scores that incorporate epidemiologic and personal risk factors [108]. Opportunities to support AYA receiving treatment for TB disease, which have been described elsewhere, also need urgent attention [22,23].

Here, we describe opportunities to expand case-finding and provision of TPT among AYA at both the community and individual levels (Figure 2). For the greatest efficacy, case-finding and prevention strategies should carefully consider the specific needs of local AYA and incorporate adolescent-friendly practices.

### 5.1. Setting-Specific Opportunities in Case-Finding and Prevention

Attention to the different settings of transmission among AYA can help guide targeted expansions of active case-finding and prevention approaches.

#### 5.1.1. Households

Systematic contact investigations should be performed in the households of all individuals diagnosed with TB disease. In these households, the WHO End TB Strategy recommends providing TPT to all asymptomatic children under age 5 years to stop progression to TB disease [18]. This high-impact approach has unfortunately proved challenging to implement and is widely underutilized [109]. AYA with household exposures were not included in early TB preventive therapy recommendations. One model of global household contact management implementation demonstrated that providing TPT to all children under age 5 years and all older household contacts with TB infection would prevent nearly 160,000 cases of TB disease and 110,000 deaths in children under age 15 years, with the largest effects seen in children under age 5 years [110]. In 2020, amid growing evidence of the increased risk to household-exposed AYA and older adults, the WHO issued updated guidance that suggests expanding TPT to all household contacts, regardless of age [18]. The implementation and impacts of this change are not yet known. Further studies are warranted to explore the best implementation strategies for and impact of expanding TPT strategies to all household-exposed AYA.

#### 5.1.2. Schools

School-based active case-finding and provision of TPT can reduce TB transmission among AYA in both low-prevalence and high-prevalence regions [87,111,112,113]. In low-prevalence areas, to limit the harms of false-positive tests for TB infection and apply resources wisely, approaches should focus on AYA at increased risk for TB [114,115]. In high-prevalence areas, universal approaches, such as a single school-entry TB screen or annual TB screening, may be warranted. Universal evaluation for TB infection and disease at the time of school enrollment significantly shortened the time to TB diagnosis and decreased the TB prevalence among AYA students in Dalian, China [113]. At schools for Tibetan refugees in India, evaluating all students also led to decreased TB prevalence [87,116]. By contrast, a more limited approach that used symptom-based screening for school contacts in Swaziland did not identify any additional AYA with TB disease [117].

School-based programs have limitations that begin foremost with the need for appropriate resources. Such programs may not reach all AYA, particularly in communities with lower levels of school enrollment [118]. Nevertheless, school-based programs can make TB care accessible to AYA at convenient times and locations, which are important features of adolescent-friendly services [4]. Investigators of an outbreak that occurred at a school in China just before national examinations noted that students’ reluctance to interrupt their studies to seek care allowed infections to spread further [85]. Similarly, AYA with TB disease in Botswana expressed unwillingness to leave school or work in order to receive treatment [23]. Access to earlier active case-finding and TPT might prevent such disruptions related to TB disease management, and school-based options for the provision of both TPT and TB treatment should be evaluated further. One school-based pilot program targeting AYA at increased risk for TB in Texas, USA, successfully provided risk screening, risk-based testing for TB infection, and school-administered directly-observed TPT [19]. Students in both low-prevalence and high-prevalence regions have viewed school-based TB screening and treatment programs favorably [112,119,120].

#### 5.1.3. Other Settings

Optimal approaches to case finding and prevention in other community settings need further study. Screening all AYA before they enter a setting with high transmission risk, such as a congregate residence, may provide benefit in some situations. Any universal approach, however, should respond to changes in local TB epidemiology. Universal screening successfully identified many AYA with untreated TB infection or disease who enrolled in the United States military, for example, but still missed some AYA and was discontinued when declining TB incidence made false-positive results more likely [102,121]. Using social media to better understand AYA social networks may also aid case-finding [122].

### 5.2. Opportunities in Case-Finding and Prevention for AYA at Increased Risk

Opportunities for improving TB case-finding and prevention among AYA begin with identifying AYA at increased risk for TB. Healthcare providers must recognize the risk factors for TB acquisition and the symptoms of TB disease in their AYA patients. Formal guidance exists for active screening in some groups of at-risk AYA, such as AYA living with HIV and those starting certain immunomodulating medication regimens [18,114]. For other at-risk groups, such as migrants and those experiencing homelessness or other vulnerabilities, screening guidance is limited, variable, and often not targeted to AYA [34,35,55,123]. Case-finding strategies should incorporate the unique needs of vulnerable AYA.

TB preventive therapy is important for AYA with immunocompromising conditions. Isoniazid preventive therapy (IPT) reduces the risk of TB disease among adults living with HIV and has been included in the WHO’s guidelines for global HIV care since 2011 [18]. For AYA living with HIV, the benefits of IPT are not well-known given limited data [124,125]. Barriers to the uptake of IPT, such as concerns about medication side effects and limited access to medications, also influence AYA living with HIV and their healthcare providers [126,127,128]. For AYA with altered immunity due to immunomodulating biologic agents (e.g., TNF-alpha inhibitors) or other reasons, no data on the use of IPT exist.

Adolescent-friendly TB services incorporating characteristics of quality adolescent-friendly health services are urgently needed to support optimal TB outcomes for AYA [4]. The WHO defines these adolescent-friendly characteristics within a framework of equitable, accessible, acceptable, appropriate, and effective elements of care to guide the development of approaches that meet all aspects of AYA needs [4]. Such approaches may include integration of peer support and of additional modes of TB education and counseling, such as via mobile apps or social media. Given the impacts of TB and its treatment on social isolation and disruption of education, optimal care for all AYA should also include close attention to their mental health needs [4]. Intentional overdose of isoniazid occurs among AYA receiving treatment for TB infection, and marginalized AYA may be at increased risk [129,130].

Educational measures should also strive to improve TB awareness among AYA and decrease the perceived stigma still attached to TB [131,132]. Even brief educational interventions may have great impact. Researchers who conducted a one-time school-based educational intervention with AYA at increased risk for TB in Texas, USA, noted that 78% of AYA reported sharing their new TB knowledge with others, thus supporting greater community-wide TB awareness [119].

## 6. Conclusions

Strategies to better prevent and treat TB among AYA are critical to halting TB transmission within communities. As we write, the pandemic caused by a novel respiratory pathogen, SARS-CoV-2, continues worldwide, with a disproportionate impact on regions with high TB prevalence. Estimates predict that disruptions triggered by the COVID-19 pandemic could cause an additional 1 million people to develop TB each year through 2025 [5]. The pandemic has further exposed numerous health systems gaps that also remain critical areas in the prevention and care of TB among AYA, from lack of equitable access to rapid diagnostics and efficient contact-tracing, to inadequate ventilation in schools and other indoor areas, to harmful effects of social isolation on AYA development. With TB cases projected to increase over the coming years, it will be critical to both increase efforts and develop more effective approaches to identify, treat, and prevent TB among AYA. Next steps should include implementing adapted and novel strategies in case-finding and treatment that recognize the specific TB risks and developmental, clinical, and mental health needs of AYA.

## Figures and Tables

**Figure 1 tropicalmed-06-00148-f001:**
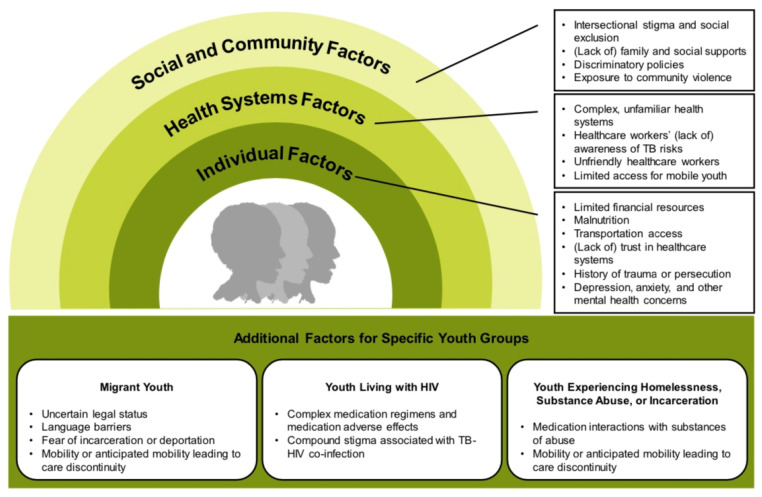
Barriers to TB Care for Marginalized Adolescents and Young Adults.

**Figure 2 tropicalmed-06-00148-f002:**
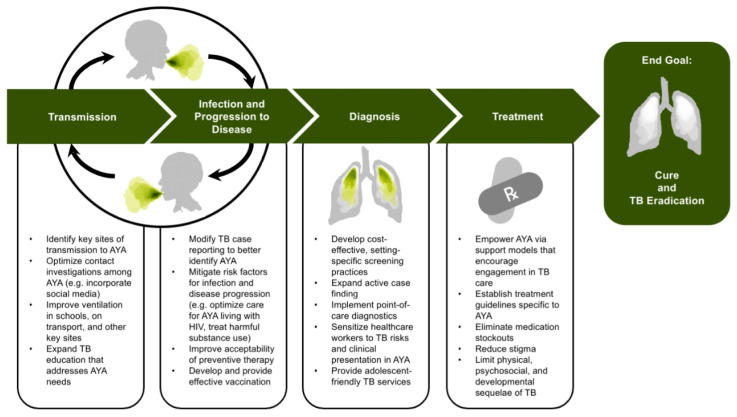
Public Health Priorities and Areas for Further Research and Implementation to Address TB among Adolescents and Young Adults.

**Table 1 tropicalmed-06-00148-t001:** Risk Factors for TB Infection and Progression to Disease in Adolescents and Young Adults.

Individual Factors	Community Factors
Malnutrition Immunosuppression (e.g., HIV infection, use of immunomodulating biologic agents) Diabetes mellitus Obesity Smoking Harmful alcohol use	Community prevalence of TB Overcrowded housing, schools, transit, and jails/prisons Air pollution

**Table 2 tropicalmed-06-00148-t002:** TB Preventive Treatment Regimens Recommended by the World Health Organization for Adolescents and Young Adults [18].

Medication (s)	Dosing Interval	Duration
*For drug-susceptible TB:*		
Isoniazid monotherapy	Daily	6 months (6H) or 9 months (9H)
Rifampicin monotherapy (4R)	Daily	4 months
Rifampicin plus Isoniazid (3HR)	Daily	3 months
Rifapentine plus Isoniazid	Weekly (3HP)	3 months (12 doses)
Rifapentine plus Isoniazid	Daily (1HP) *	1 month (28 doses)
*For multidrug-resistant (MDR) TB:*		
Levofloxacin	Daily	6 months

* This regimen is recommended only for ages 13 years and older, as daily dosing of rifapentine has not yet been established in children and adolescents under age 13 years [18].

## Data Availability

No new data were created or analyzed in this study. Data sharing is not applicable to this article.

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
