# Peer review of "Tuberculosis in Adolescents and Young Adults: Emerging Data on TB Transmission and Prevention among Vulnerable Young People"

_tropicalmed, 2021, doi:10.3390/tropicalmed6030148_

Round 1

Reviewer 1 Report

I read with interest this paper that focuses on the important issue of  B risks, prevention, and care for vulnerable young people. I congratulate the authors for providing a sound contribution 

Author Response

Thank you very much for taking the time to review our manuscript. We appreciate your kind feedback.

Reviewer 2 Report

This is a welcome paper because I think it is good to focus on AYA (as officially, we only care about aged below 15 and above 15 years). However, I have two major points that need attention.

Major points:

The methods are not clear. It is clear that this was not a systematic review. So what were the criteria for the selection of the documents that were consulted (and listed in the reference section)?  

Important information is not or only scarcely available: latent TB and active TB in the various age groups. See https://tbrieder.org/ as a good source. For instance, what are the risks of developing active TB once infected among the various age groups. This is a key piece of information. E.g. in Line 46, the 10% is mentioned. But this percentage depends on age groups!

Other points:

References are used without critically looking at them. Example: reference 95 (Aksenova et al): The end of the abstract reads: “Annual screening for TB exposure with treatment for latent or active TB has reduced the annual incidence of TB in Russian children aged 15–17 years to 1992 levels.” This cannot be stated the way it is; there are multiple reasons for the decline and it is very doubtful that the annual screening contributed substantially. Also, in this paper by Aksenova, the age groups vary, making interpretations problematic.:  

As TB preventive therapy, isoniazid is mentioned. But other regimens (e.g. with rifapentine) are not mentioned. A brief mention would be warranted, also because especially with AYA, adherence to 6 to 9 months treatment is problematic. With a 2-month regimen, this challenge is less grave.

Minor points:

Line 16: now COVID-19 is leading.

Table 1: I think jail/prison/inmates should also be mentioned.

Lines 106-109: this is repetitive.

Line 109: not only AYA but TB patients in general with incomplete treatment!

Line 121: migrants: you may want to refer to the study of Walker et al, Lancet Inf Dis.: Lancet Infect Dis 2018, 18: 431–40; http://dx.doi.org/10.1016/S1473-3099(18)30004-5. These 29 migrants all were younger than 26 years old.

Line 141: reference needed at the end.

Line 164: the word “to” is missing.

Line 174 mentions the “the typically longer interval between TB infection and onset of TB disease”. Can you elaborate, presenting figures?

Lines 188-9: (ref. 55): it seems the findings from children aged 1-9 yrs are missing.

Line 297: here, stress is mentioned. However, it would be good to also mention stress/psychological issues as a risk factor in the sector on around line 125.

Figure 2: the title has “Research Gaps”. However, these are not presented. It may be good to have for all four columns at bottom an additional box with the research gaps there.

Line 338: I suggest implementation first, then impact, thus: “explore the best implementation strategies for and impact of…”

Reviewer 3 Report

This is an outstanding review paper focusing on tuberculosis in adolescents and young adults (AYA). A difficult topic with interconnected social and epidemiological factors, but very well introduced, described and analyzed by the authors. The article highlights possible strategies to better promote diagnosis and treatment for this particular population, which are likely to be useful in the context of other respiratory infectious pathogens, like SARS-CoV-2.

I have only three minor comments:

  • line 46, it is written: "Individuals with TB infection (...) cannot transmit TB to others". I would tone this down, because asymptomatic individuals are still at risk to transmit the disease, although this is not well documented. Maybe "are at low risk of transmitting" or "unlikely to transmit"
  • line 90, prevalence of diabetes mellitus and line 105, high rates of loss to follow up: it would be nice to indicate these stats in the text, ideally including a comparison with adult group
  • Figure 1 is truncated in my document

Reviewer 4 Report

Thanks for the opportunity to review this great article on adolescent TB. The review is timely, is well written, and has a lot of great information; I fully support its publication. I have some suggestions for improvement below.

MAJOR COMMENTS:

Page 2, Lines 75-78: I suggest wording these lines a bit more cautiously. While we think that the immunological changes associated with puberty are driving the changes in TB risk and clinical phenotype, we don’t know for sure.

Page 2, Lines 81-86: This section could be strengthened with more nuance. During early adolescence, the risk is equal between boys and girls. In mid-adolescence, the risk increases in girls; this pattern has been shown in both literature from the mid-20th century (Edith Lincoln observed the that risk goes up around the time of menarche; see the Seddon JA, et al. Front Immunol paper for other citations) and more current epidemiologic studies (the Chiang SS, et al. ERJ Open article that you cited from Ukraine). However, in later adolescence, the risk in boys overtakes the risk in girls, and this imbalance remains during most of adulthood, as you have already mentioned. Another point worth mentioning is the high HIV risk among girls and young women aged 15-24 years, and how this risk impacts TB risk. I am also curious if the differences observed in the Cape Town study reflect different HIV prevalence among the older adolescent boys vs. girls?

Part 4: This section seemed the weakest to me. I think the title could be improved because, if I’m understanding correctly, it’s supposed to be about identifying AND treating both TB infection and disease in AYA? The section about treating TB disease is pretty scant (I believe it starts on Page 10, line 395) and could be expanded. Important topics including attention to mental health are just mentioned in passing.

MINOR COMMENTS:

Page 1

Lines 16-17 and 29: Technically, right now, Covid-19 is currently the leading single-agent infectious case of death worldwide. I think adding “single-agent” is also important because pneumonia and diarrhea (from multiple different pathogens, including M. tuberculosis) cause more death than TB disease.

Page 3

Lines 105-106: What do you mean by other unfavorable outcomes. Generally, adolescents experience a lower risk of death from TB than adults.

Page 4

Figure 1: I cannot see all of the figure; it’s cut off on the right-hand side.

Lines 129-130: Can you briefly specify what additional challenges these AYA living in cross-border regions face?

Page 8

Figure 2: “Provide effective vaccination” . . . I think it’s better to say “Develop and provide effective vaccination” or something like that because just saying “provide” may give the false impression that we already have a good TB vaccine

Figure 2: “Develop cost-effective, country-specific screening practices” . . . I think “setting-specific” would be better because there are many differences within countries (e.g. , capital city vs. rural areas)

Page 9:

First paragraph: Just an idea: it may be interesting to mention ongoing work to develop and validate risk scores based on epidemiological, clinical, and/or immunological factors to target the highest-risk AYA (e.g., Rishi Gupta’s study in Nature Med 2020).

Section on schools: Another pilot program to educate and screen adolescents in schools that is worth mentioning is described here: Hatzenbuehler LA, et al. Pediatr Infect Dis J 2016;35:733-738 and Hatzenbuehler LA et al. Patient Educ Couns 2017;100:950-956.

Round 2

Reviewer 2 Report

My comments have been taken on board well. I have only a few suggestions of further improvement. 

  1. You adjusted the text and now have: "We strove to include only titles that provided disaggregated data for individuals between 10 and 24 years of age.". My reaction: this does not seem fully clear to me. Did you only look at titles that mentioned the age group 10 to 24 (or e.g. 10 to 14 and 15 to 24)? Also, the word “strove” indicates that it was not always possible. It may be good to elaborate on this.
  2. Regarding the section starting with: "In settings with high TB incidence, the prevalence of TB infection increases throughout adolescence as individuals....". My reaction: For the whole section here: please add the relevant references at various sentences of this section. 
  3. Regarding the passage: "TB disease developed in 192,000 young adolescents aged 10-14 years, 535,000 older adolescents aged 15-19 years, and 1,049,000 young adults aged 20-24 years.”: my reaction: I assume this is GLOBALLY. If so, add the word “globally” or “worldwide”. 
  4. As to Table 2: please add the reference (which WHO document?)
